# Teleconsultations between Patients and Healthcare Professionals in Primary Care in Catalonia: The Evaluation of Text Classification Algorithms Using Supervised Machine Learning

**DOI:** 10.3390/ijerph17031093

**Published:** 2020-02-09

**Authors:** Francesc López Seguí, Ricardo Ander Egg Aguilar, Gabriel de Maeztu, Anna García-Altés, Francesc García Cuyàs, Sandra Walsh, Marta Sagarra Castro, Josep Vidal-Alaball

**Affiliations:** 1TIC Salut Social—Ministry of Health, 08028 Barcelona, Spain; flopez@ticsalutsocial.cat; 2CRES&CEXS—Pompeu Fabra University, 08003 Barcelona, Spain; 3Faculty of Medicine, Barcelona University, 08036 Barcelona, Spain; ricardo.anderegg@gmail.com; 4IOMED Medical Solutions, 08041 Barcelona, Spain; gabriel.maeztu@iomed.es; 5Agency for Healthcare Quality and Evaluation of Catalonia (AQuAS), Catalan Ministry of Health, 08005 Barcelona, Spain; agarciaaltes@gencat.cat; 6Sant Joan de Déu Hospital, Catalan Ministry of Health, 08950 Barcelona, Spain; 31557fgc@gmail.com; 7Institut de Biologia Evolutiva (UPF-CSIC), Pompeu Fabra University, 08003 Barcelona, Spain; sandra.walsh34@gmail.com; 8Centre d’Atenció Primària Capellades, Gerència Territorial de la Catalunya Central, Institut Català de la Salut, 08786 Sant Fruitós de Bages, Spain; msagarra.cc.ics@gencat.cat; 9Health Promotion in Rural Areas Research Group, Gerència Territorial de la Catalunya Central, Institut Català de la Salut, 08272 Sant Fruitós de Bages, Spain; 10Unitat de Suport a la Recerca de la Catalunya Central, Fundació Institut Universitari per a la recerca a l’Atenció Primària de Salut Jordi Gol i Gurina, 08272 Sant Fruitós de Bages, Spain

**Keywords:** machine learning, teleconsultation, primary care, remote consultation, classification

## Abstract

*Background*: The primary care service in Catalonia has operated an asynchronous teleconsulting service between GPs and patients since 2015 (eConsulta), which has generated some 500,000 messages. New developments in big data analysis tools, particularly those involving natural language, can be used to accurately and systematically evaluate the impact of the service. *Objective*: The study was intended to assess the predictive potential of eConsulta messages through different combinations of vector representation of text and machine learning algorithms and to evaluate their performance. *Methodology*: Twenty machine learning algorithms (based on five types of algorithms and four text representation techniques) were trained using a sample of 3559 messages (169,102 words) corresponding to 2268 teleconsultations (1.57 messages per teleconsultation) in order to predict the three variables of interest (avoiding the need for a face-to-face visit, increased demand and type of use of the teleconsultation). The performance of the various combinations was measured in terms of precision, sensitivity, F-value and the ROC curve. *Results*: The best-trained algorithms are generally effective, proving themselves to be more robust when approximating the two binary variables “avoiding the need of a face-to-face visit” and “increased demand” (precision = 0.98 and 0.97, respectively) rather than the variable “type of query” (precision = 0.48). *Conclusion*: To the best of our knowledge, this study is the first to investigate a machine learning strategy for text classification using primary care teleconsultation datasets. The study illustrates the possible capacities of text analysis using artificial intelligence. The development of a robust text classification tool could be feasible by validating it with more data, making it potentially more useful for decision support for health professionals.

## 1. Introduction

eConsulta is an asynchronous teleconsultation service between patients and GPs as part of the electronic health records of the public primary healthcare system of Catalonia. In operation since the end of 2015, this secure messaging service was designed to complement face-to-face consultations with primary healthcare teams (PHT). It was gradually implemented up until 2017, when the service became available to every PHT; currently, all of them have used this tool at least once.

An earlier study analysed the reasons why patients sought a consultation, which resulted in a patient–doctor interaction, as well as the subjective perception of the GP if they avoided a face-to-face visit or if it led to a consultation which otherwise would not have occurred, by means of a retrospective review of text messages relating to each case [1]. The results show there was a broad consensus among GPs that eConsulta has the potential to resolve patient queries (avoiding the need for a face-to-face visit in 88% of cases) for every type of consultation. In addition, GPs declared that ease of access led to an increase in demand (queries which otherwise would not have been made) in 28% of cases. Therefore, the possibility of eConsulta replacing a conventional appointment stands at between 88% and 63% (88% × (1 − 28%)). The most common use of e-consultation was for the management of test results (35%), clinical enquiries (16%) and the management of repeat prescriptions (12%).

Technology offers new possibilities for policy evaluation in conjunction with the aforementioned classical approaches. Artificial intelligence tools are already widely used in the field of healthcare in areas such as the prediction and management of depression, voice recognition for people with speech impediments, the detection of changes in the biopsychosocial status of patients with multiple morbidities, stress control, the treatment of phantom limb pain, smoking cessation, personalized nutrition by prediction of glycaemic response, to try to detect signs of depression and in particular for reading medical images [2,3,4,5,6]. The generation of data implies a huge potential for the impact assessment of these interventions with new analytical tools.

The classification of texts in the medical field has also been used to conduct a review of influenza detection and prediction through social networking sites [7,8,9] and in the analysis of texts from internet forums [10,11]. More specifically, in the framework of teleconsultations, a US-based study used machine learning to annotate 3000 secure message threads involving patients with diabetes and clinical teams according to whether they contained patient-reported hypoglycaemia incidents [12]. As far as the authors are aware, no study has looked into the development of a text classification algorithm in the context of teleconsultations between patients and primary care physicians.

The present study aims to evaluate specific text classification algorithms for eConsulta messages and to validate their predictive potential. The algorithms have been trained using a vector representation of text from the body of the message and the three variable annotations that primary healthcare professionals in Central Catalonia used in a previous study: avoiding the need for a face-to-face visit, increased demand and type of use of the teleconsultation [1]. Our study represents an exhaustive exploratory analysis of text classification algorithms of teleconsultation messages between GPs and patients that can provide useful information for future research and a potential use for decision support in healthcare.

## 2. Methodology

### 2.1. Data Acquisition

The teleconsultations that had previously been classified that were used as the basis for training the algorithm are those which were acquired in the study by a previous study (López) (Table 1). They are part of the health records of the *Gerència Territorial de la Catalunya Central* of the *Institut Català de la Salut* covering the period from when the tool was first used until the date of its extraction for analysis purposes (8 April 2016 to 18 August 2018). Message deidentification was performed by substituting all possible names contained in the Statistical Institute of Catalonia database [13] with a common token and removing all other personal attributes. The classification method used for the conversations is described and justified by López et al. 2019: Every healthcare professional who received an eConsulta labelled it according to whether, in their opinion, it avoided the need for a face-to-face consultation, led to an increased demand and by type of teleconsultation (Section A.1). These results of this annotation, with the corresponding messages, were used to train the text classification model using the three variables previously mentioned (Table 2).

Most of the data were received with a tabular arrangement, and the texts and their labels were in different files that were merged according to the Conversation ID. The data cleaning was a multi-step process. Regarding the text: First, all the tokens of anonymized names were changed to a standard name of the country “Juan”. The title was merged with the body of the message, adding the token “xxti” before the title and “tixx” after the title; that way we would not lose the information that this was the title. The texts were all converted to lowercase, and we extracted the length (in words and in characters) of every message to use as extra independent variables. As additional variables, the day of the month and time of the day were extracted from the date of the message.

### 2.2. Vector Representation of Text in eConsulta Messages

The emails needed to be represented in some way in order to use them as input for the models. A common practice in machine learning is the vector representation of words. These vectors capture hidden information about the language, such as word analogies and semantics, and improve the performance of text classifiers.

Four techniques have been used to generate the vector representation of texts. The Bag of Words (BoW) approach counts the number of times pairs of words appear in each document. The document is represented as a vector of a finite vocabulary. The Term Frequency–Inverse Document Frequency (TF–IDF) method assigns paired words a weight depending on the number of times they appear in a particular document (the Term Frequency), while discounting its frequency in other documents (Inverse Document Frequency): The more documents a word appears in, the less valuable that word is as a signal to differentiate any given document. Word2Vec is a two-layered neuronal network that trains and processes text. Its input is a corpus of text and its output is a set of vectors for the words in the corpus, with words represented by numbers. The initial vector assigned to a word cannot be used to accurately predict its context, meaning its components must be adjusted (trained) through the contexts in which they are found. In this way, repeating the process for each word, word vectors with similar contexts end up in nearby vector spaces. Fasttext [14] is used to obtain word2vec vectors. Finally, the objective of Doc2vec is to create a numerical representation of a document, regardless of its length. This approach represents each document by a dense vector, which learns to predict the words in the document [15]. In all cases, before carrying out the vectorization of the texts, these were first tokenized and any stop-words eliminated (those which are taken to have no meaning in their own right, such as articles, pronouns or prepositions).

In each instance, the vectors were enriched by supplementing them with similar texts in Catalan and Spanish [16]. The external data used to enrich the corpus were models of interactions extracted from online databases with colloquial language similar to that used in eConsulta. Where augmented BOW, TF-IDF and Word2Vec were used, word and character length and word density were also used as predictor variables.

### 2.3. Training and Testing AI Algorithms

The task addressed in this study is a multiclass classification with respect to the type of visit and two binary classifications for the other two variables (avoiding visit and increased demand). For each text vector representation algorithm five different algorithms were implemented: Random Forest, Gradient Boosting (lightGBM), Fasttext, Multinomial Naive Bayes and Naive Bayes Complement [17]. Bayesian text classifiers are the most standard algorithms in this setting. A convolutional neural network was also used using the augmented Word2vec vectors. We tested the performance of the algorithms through a stratified 10-fold cross-validation: During 10 iterations/trainings, 9 divisions served as learning and 1 as a test.

The coefficients of interest to evaluate the goodness of the algorithms were precision (the fraction of relevant instances between the retrieved instances/proportion of correct predictions of the total of all predicted cases) and sensitivity (the number of correct classifications for the positive class “true positive”). It was decided not to use the “accuracy” coefficient since it is a metric that, given an unbalanced dataset like the one under investigation, can result in a very high score in spite of the fact that the classifier works poorly, since it assesses the number of total hits without taking into account whether most of the data is of the same class. The F value is used to determine a weighted single value of accuracy and completeness. The diagnostic value is assessed by means of the ROC curve. The goodness-of-fit of all the coefficients is represented as a value between 0 and 1.

Python 3.7 and the following libraries were used for the algorithm training: numpy [18], matplotlib [19], seaborn [20], altair [21], scikit-learn [22], pandas [23], gensim [24], nltk [25], fasttext [14], pytorch [26] and lightGBM [27]. The majority of the code was carried out on Jupyter Notebooks [28].

### 2.4. Ethical Considerations

The study was approved by the Ethical Committee for Clinical Research at the Foundation University Institute for Primary Health Care Research Jordi Gol and Gurina, registration number P19/096-P, and carried out in accordance with the Declaration of Helsinki [29].

## 3. Results

In order to assess the predictive potential of eConsulta messages regarding the three variables of interest, we first aimed to identify the best combination of algorithms. A total of 3559 messages (169,102 words) corresponding to 2268 teleconsultations (1.57 messages per teleconsultation) were analysed in a framework of 20 different combinations of vector representation of text and machine learning algorithms (Table 3). We assessed the performance of the combinations of algorithms though a stratified 10-fold cross-validation analysis. Figure 1 shows the performance of the most stable algorithm (best metrics, in general) according to the predictor variable.

Specific combinations of algorithms per variable generally perform very well. Table 4 shows the evaluation metrics (mean + standard deviation of the 10 iterations) of the combination of algorithm and numerical representation of the text which has a better performance for each target variable. For all of the cases, the vectors obtained directly from the original texts have been more useful than those enriched with external texts. Table 4 shows that algorithms are generally effective, showing they are better when approximating the two binary variables (avoiding the need for a face-to-face visit, increased demand) than the variable “type of query”. Thus, eConsulta’s classifiers have a promising and robust predictive value, especially for binary variables.

As a whole, the results illustrate eConsulta’s algorithm classifiers potential predictive value and provide a valuable insight into the implementation of AI methodologies for healthcare teleconsultation.

## 4. Discussion

### Limitations

Several limitations apply to this study and the results must be understood in light of these shortcomings. First, our classifier is restricted to one dataset and the training set was relatively small. Although the study used all the available information, more data is needed to generalize the model and avoid overfitting.

The amount of data with which the algorithms were tested is especially relevant in the case of trying to calculate the variable “type of message”, since the number of types which contain the classification [13], meaning the quantity of messages of each with which the classification algorithm has been trained, is minimal, thus diminishing its predictive capacity. This may have had implications to our approach and subsequent results. What is required is not only more messages, they must also contain as much information as possible. Validating the algorithm requires a replication of the proposed methodology with a larger data set, together with the analysis of subgroups. Likewise, the goodness of fit of the results may be caused by overfitting: The model explains this set of data well, but could show weaknesses when generalizing to others, limiting its potential for extrapolation. Because of that, this study includes exhaustive detail of the methodology used in order that it can be replicated.

Second, an error analysis was not conducted. This analysis might have helped us to understand why certain posts where misclassified or classified correctly.

Using complex mathematical models makes it difficult to explain why some work better than others. The vectors would need to be evaluated at a lower level in order to have a better idea as to which characteristics redirect the model towards one decision or another. This analysis is of interest for future applications of these techniques on a larger scale or for applications related to medical practice.

## 5. Conclusions

In Catalonia, the number of conversations and messages now stand at approximately 370,000 and 500,000, respectively. Applying a classification algorithm like the one proposed here would help us understand the nature of the conversations and their impact in real time. Future research should evaluate the use of automation (to send a diagnostic test, generate an alert or “thank you” and close the case) as a tool for decision support for healthcare professionals to improve the management of clinical cases and to save GPs time. Natural Language Processing approaches should further analyse the content of the teleconsultations and proactively offer clinicians agile resources to deal with the cases.

This article has shown that the implementation of an algorithm for the prediction of factors such as a reduction in the number of face-to-face visits, induced demand or type of consultation is technically feasible and potentially useful in the context of service planning, management of the demand and evaluation. This study presents a combination of algorithms based on machine learning and a more efficient representation of vectors for this type of data. This study is an initial exploration into the potential of teleconsultation and the promising use of artificial intelligence for the evaluation of digital health interventions.

## Figures and Tables

**Figure 1 ijerph-17-01093-f001:**
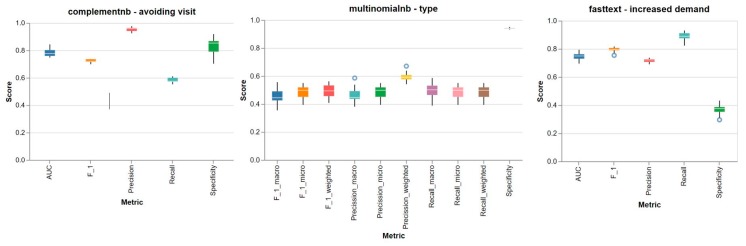
Performance metrics of algorithms.

**Table 1 ijerph-17-01093-t001:** Data recorded by the teleconsulting system.

Conversation Title	Conversation ID	Message ID	From	To	Message	Files Attached?
Travelling to Australia	C1	M1	Mr. John Patient	Ms. Jane Doctor	Dear doctor, I’m travelling to Australia on 15 December. Do I need to have any vaccinations? Many thanks	No
		M2	Ms. Jane Doctor	Mr. John Patient	Hi, vaccination is required for travel to Australia	No

**Table 2 ijerph-17-01093-t002:** Annotation by the GP.

Conversation ID	Face-to-Face Visit Avoided?	Increased Demand?	Type of Visit
C1	Yes	No	6 (Vaccinations)

**Table 3 ijerph-17-01093-t003:** Text representations and algorithms used.

Text Representations	Algorithms
BoW	Random Forest
TF–IDF	Gradient Boosting (lightGBM)
Word2Vec	Fasttext
Doc2Vec	Multinomial Naive Bayes
	Complement Naive Bayes

**Table 4 ijerph-17-01093-t004:** Results of the best algorithm/text representation combination, according to the variable to be approximated. Average (SD) of the 10 iterations.

Variable	Precision	Recall	F1	Roc_AUC
Avoiding the need of a face-to-face visit	Random ForestTF-IDF0.98 (0.026)	FastTextWord2Vec0.99 (0.005)	FastTextWord2Vec0.92 (0.004)	ComplementNBTF-IDF0.79 (0.032)
Increased demand	Random ForestTF-IDF0.97 (0.057)	FastTextWord2Vec0.89 (0.029)	FastTextWord2Vec0.79 (0.018)	FastTextWord2Vec0.75 (0.031)
Type of use of the teleconsultation(micro averaged score)	MultinomialNBBOW0.48 (0.049)	MultinomialNBBOW0.48 (0.049)	MultinomialNBBOW0.48 (0.049)

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
