# Peer review of "Teleconsultations between Patients and Healthcare Professionals in Primary Care in Catalonia: The Evaluation of Text Classification Algorithms Using Supervised Machine Learning"

_ijerph, 2020, doi:10.3390/ijerph17031093_

Round 1

Reviewer 1 Report

The article is generally well written. The topic is of interest, as indeed there is a limited number of studies on the topic in the European Countries. However, it needs a minor revision before can be recommended for publication. The authors should re-write the references section according to the journal's instructions for authors.

Author Response

Dear reviewer 1,

Thanks a lot for your comments. We have attached a file which has taken into account yours and other reviewers’ suggestions.

The references have been adjusted according to the "MDPI Reference List and Citations Style Guide".

Please take into account that some of the references are related to software packages, which are not published in standard journals and thus cited in a non-standard manner.

Best regards,

Francesc López Seguí

Reviewer 2 Report

Great article with a good focus and the discussion is well written. A major pain point in digital healthcare is to how to encourage non-IT professionals to understand and thus gain interest in such initiatives. With that, I strongly suggest the authors to define and explain some of the common data science terms. One example will be the algorithms used in table 3. 

Last, for the total of 3559 messages and its relevant data obtained, did the authors perform data cleaning? It will be interesting for the readers to understand the challenges in data mining and accuracy.   

Overall, this is a neat article that will benefit the digital healthcare community. 

Good Job!

Author Response

Dear reviewer 2,

Thanks a lot for your comments. We have attached a file which has taken into account yours and other reviewers’ suggestions.

It is very difficult to explain technically, using plain language, how do the text representations and algorithms work. Despite that, this is extensively done in sections 2.2 and 2.3. However, an algorithm is somehow a "blackbox". This is also explained in the discussion.

A paragraph explaining the challenges related to the data cleaning has also been included.

Best regards,

Francesc López Seguí

Reviewer 3 Report

I think your study is good in itself, but I would like to see it linked to a greater discussion, a research gap and theoretical framework. The way it stands now, it reads as a technology report and it doesn't give us much more information than that the best- trained - algorithms were effective. Could you link this to a greater debate and perhaps a theoretical framework?

Author Response

Dear reviewer 3,

Thanks a lot for your comments. We have attached a file which has taken into account yours and other reviewers’ suggestions.

We have rewritten the conclusions section to further motivate the interest of this study. We are discussing the utility of classification algorithms in service planning, management of the demand and evaluation. There are potential applications in automation (to send a diagnostic test, generate an alert or "thank you" and close the case). We are mentioning these topics, although it is out of the scope of the article as we cannot go further / prove them. The introduction also defends the idea of the use of technology to analyse digital health: Artificial intelligence to evaluate the interventions. Using technology to analyse technology.

The main idea derived from the article is indeed that text classification algorithms are feasible and could be used in a clinical setting if the classifiers are validated with a bigger dataset. This is the reason why the article is called "evaluation of text classification algorithms". We have consciously tried not to derive further conclusions because the training set is relatively small. This article adds to other (pioneer) experiences in the use of technology for the evaluation of digital health.

Best regards,

Francesc López Seguí
